# Respiratory Syncytial Virus Infection in Children with Acute Lymphoblastic Leukemia (ALL): A Contemporary Emerging and Struggling Clinical Event

**DOI:** 10.3390/pediatric17050095

**Published:** 2025-09-17

**Authors:** Marta Arrabito, Emanuela Cannata, Luca Lo Nigro

**Affiliations:** 1Center of Pediatric Hematology Oncology—Azienda Policlinico, 95123 Catania, Italy; lucalonigro1968@gmail.com; 2Department of Translational Medicine, University of Catania, 95131 Catania, Italy; e.cannata80@gmail.com

**Keywords:** respiratory syncytial virus, acute lymphoblastic leukemia, children, airways infection

## Abstract

Systemic viral infections are frequently life-threatening in immunocompromised children. Many viral pathogens are reported to be the cause of morbidity and mortality in these pediatric patients, but scarce evidence is related to respiratory syncytial virus infection (RSV), which is one of the main viral causes of lower respiratory tract infection in infants and young children. Herein we report the experience of the Center of Pediatric Hematology Oncology of Catania regarding RSV infection in pediatric leukemia patients, describing four cases: three with only respiratory involvement and complete recovery (two of them presented mild symptoms and one evolved into severe respiratory failure) and a fourth case with an initial hepatic and pulmonary involvement leading to death. Unfortunately, some viral infections have delayed diagnoses because of lack of awareness and atypical presentation. Therefore, our intent is to highlight the importance of mindfulness of the occurrence of this infection and of its typical and atypical manifestations in order to detect it early and decrease the risk of morbidity and mortality.

## 1. Introduction

Systemic viral infections are frequently life-threatening in immunocompromised children. Many viral pathogens are reported to be the cause of morbidity and mortality in this type of pediatric patients, but scant evidence is related to RSV, even though it is estimated that 33 million episodes of these specific infections occur in children younger than five years, resulting in 3.6 million hospitalizations and 118,200 deaths [1]. RSV is a single-stranded ribonucleic acid virus belonging to the Paramyxoviridae family [2] and one of the main viral causes of lower respiratory tract infection in infants and young children. During this infection, RSV causes epithelial cell sloughing, cilia loss, sporadic syncytial body formation, and excessive mucus secretion. The destruction of these epithelial cells commonly leads to discontinuity, with a small group of epithelial cells infected in the respiratory tract. The loss of cilia is usually associated with microtubule damage, which occurs as the virus replicates in the apical cell surface. This detached cellular debris reaches the narrow-diameter bronchiolar airway lumen, inducing the accumulation of debris and acute obstruction in the distal airways [3]. A total of ≥40% of children report nasal discharge/congestion, cough, shortness of breath, feeding abnormalities, and fever [4]. The clinical manifestations of pediatric RSV infection may vary according to age. Infants and young children could develop potentially life-threatening low respiratory tract infections manifesting as bronchiolitis and/or pneumonia, whereas older children usually present with mild upper respiratory tract infections [5].

In the majority of the pediatric population, the course of the infection is benign, but in certain groups of high-risk patients, such as premature infants or children with malignancies undergoing chemotherapy or after allogeneic hematopoietic stem cell transplantation (Allo-HSCT), RSV is one of the most lethal pathogens [6]. Indeed, acute leukemia is a disease of the immune system and, therefore, presents greater susceptibility to infections. This study reports the experiences at our center regarding RSV infection in pediatric patients with acute lymphoblastic leukemia (ALL), describing one case manifested mainly with fatal hepatic and pulmonary involvements and three cases presented with mild or severe respiratory failure. We included patients who developed RSV infection associated with ALL diagnosed at our center from the COVID-19 era of 2020 to the present.

## 2. Case Presentations

### 2.1. Case 1 [Unique Patient Number (UPN) 1069066]

A three-year-old female received a diagnosis of B-cell precursor (BCP)-ALL, bearing the t(12;21) and was enrolled in AIEOP-BFM ALL 2017 protocol.

The evaluation of bone marrow aspirate (BMA) performed at day +15 (d + 15) showed a 0.004% lymphoblast with BCP-ALL immunophenotype, compatible with standard risk. Due to the contemporary occurrence of fever and serous rhinorrhea, the planned chemotherapy of d + 15 (vincristine and daunorubicin) was delayed and an antibiotic treatment with cephalosporin was started. At chest radiography, a thickening of the peribronchial interstitium was detected. After two days, the fever disappeared and the planned dose of vincristine and daunorubicin was administered.

In the following days, the patient presented fever, rhinorrhea and dry cough, maintaining normal vital signs with a mild increase in inflammatory index. Therefore, the antibiotic treatment was modified by adding an anti-Gram-positive bacteria drug.

On days 21 and 22 of the induction phase, the child still presented with fever, pancytopenia, and elevated liver enzyme (aspartate aminotransferase-AST) (see Table 1). Thus, the patient underwent a computed tomography (CT) of the chest, which highlighted small areas of pulmonary consolidation of triangular morphology in the bilateral posterobasal mantle area and a reduction in the diaphony of the lung parenchyma (Figure 1A), compatible with a viral infection. Consequently, due to the pandemic period, a swab for viral infection SARS-CoV-2 was performed but had a negative result. Two days after, a broncho-alveolar lavage was performed to test many viruses, such as A & B influence, SARS-CoV-2, and RSV. In the meantime, the transaminases values gradually increased, showing that the AST value was higher than alanine aminotransferase (ALT). Moreover, an alteration of coagulation (Table 1) leading to a plasma infusion was detected.

On d + 23, pancytopenia persisted, and liver enzymes were dramatically increasing with a mild increase in the total bilirubin (Table 1). The fever was still present, and the child’s general conditions were poor: she manifested abdominal pain treated with morphine and difficult breathing faced with high O2 flow in the nasal cannula. On Day +24 the general conditions advanced, and the child presented with numb sensory, tachycardia, tachypnea, contracted diuresis with persisting high transaminase, and coagulation values were not measurable. A cardiac arrest occurred, and cardiopulmonary resuscitation was performed by intubating the patient and administering adrenaline. A nasogastric tube was inserted, and fluid therapy was administered correcting bicarbonates and glucose.

The patient was stabilized, but after a few hours, she presented with severe hemorrhagic events from the respiratory tract. Her systolic pressure decreased and therapy with noradrenalin was started. An urgent CT chest–abdomen was performed which showed massive pleural effusion with lower lobe atelectasis in the chest district, and inhomogeneous liver parenchyma with inflammatory involvement (Figure 1B). There were intestinal intraluminal blood spills, abundant abdominal effusion, and multiple triangular hypodense corticomedullary areas in both kidneys, such as pyelonephritis. Subsequently, the child was moved into the intensive care unit (ICU) where she died a few hours later. The broncho-alveolar lavage had positive results for high copies of RSV and negative for other viruses, bacteria, and fungi.

### 2.2. Case 2 (UPN 1077926)

A ten-year-old male received a diagnosis of Early T-cell precursor (ETP)-ALL and was enrolled in AIEOP-BFM ALL 2017 protocol. In his clinical history, he presented recurrent respiratory symptoms, such as cough and bronchospasm treated with inhalation therapy. The evaluation of d + 15 BMA showed 80% lymphoblasts with ETP-ALL immunophenotype, compatible with high risk. During the induction protocol IA, he presented with SARS-CoV-2 and H1N1 infections leading to a prolonged hospitalization, but without any side effects. During the induction protocol IB, he was admitted to our center for febrile neutropenia. A polymerase chain reaction (PCR) analysis using a respiratory swab was performed, which showed positivity for RSV. In particular, respiratory pathogens were detected using the BioFire^®^ FilmArray^®^ Respiratory Panel (bioMérieux, Marcy-l’Étoile, France), a multiplex PCR-based system that simultaneously identifies multiple respiratory viruses and bacteria from a single nasopharyngeal swab. The system provides automated extraction, amplification, and detection, and delivers results within approximately one hour. At the hospital, he presented with high fever (40 °C) and mild respiratory involvement (92–93% oxygen saturation (OS)). Therefore, an antibiotic therapy with cephalosporine and an oxygen support with nasal cannulas was started. Furthermore, he underwent a chest X-ray and subsequently a chest CT, which showed multiple parenchymal thickenings with a partly consolidative and partly ground glass appearance in both lungs and diffuse thickening of the peribronchovascular interstitium (Figure 2A). Because of the worsening of his general conditions with the persistence of fever and desaturation episodes (84–85% OS) leading to severe respiratory failure, he was supported with high flow nasal cannula (HFNC) and then with continuous positive airway pressure (CPAP) and non-invasive ventilation (NIV). Drug therapy was modified by adding systemic steroid and granulocyte colony-stimulating factor (G-CSF) and by replacing cephalosporin with meropenem, even when the blood cultures were negative. Furthermore, immunoglobulin infusion and blood and platelet transfusions were administered as support therapy.

After one week of therapy, general conditions improved, and the chest X-ray performed after eight days of treatment showed disappearing of the previous lesions (Figure 2B). Thus, he was gradually weaned from respiratory support and antibiotic therapy.

Currently, the child is in good clinical condition, and he is continuing the chemotherapy according to the AIEOP-BFM 2017 protocol.

### 2.3. Cases 3 (UPN 1076952) and 4 (UPN 1077940)

The last two cases are reported together because the presentation and symptoms were similar. A five-year-old male was diagnosed with ALL-B lineage, bearing the t(12;21), classified by the post-induction MRD evaluation as standard risk. He presented with the RSV infection during the reinduction phase (or Protocol II).

A six-year-old male received the diagnosis of ALL-B lineage; he was assigned to the high-risk group because of the high rate of blasts (24%) detected at d + 15 BMA. He presented with the RSV infection during the protocol IB.

They were almost simultaneously hospitalized because of febrile neutropenia, cough, and mild respiratory involvement (95–96% OS) without any respiratory support necessity. They showed positivity for RVS in the BioFire^®^ FilmArray^®^ Respiratory Panel (encompassing many viruses, such as adenovirus, coronavirus, rhinovirus, enterovirus, metapneumovirus, influenza, parainfluenza, and RSV) and the chest X-ray detected a slight accentuation of the peribronchovascular texture (Figure 3A,B). Treatment with cephalosporine and inhalator therapy were started, and preparations of immunoglobulin were used because of a decreased immunoglobulin concentration. After a few days the general conditions improved, and both patients recovered, enabling them to continue the chemotherapy protocol.

## 3. Discussion

Respiratory diseases are a source of morbidity and mortality in childhood, especially for immunocompromised children as patients undergoing chemotherapy. In particular, evidence shows that ALL is the most prevalent neoplasia in which respiratory infections occur [7]. While most of these infections are not lethal in healthy children, they can be life-threatening in ALL patients. Many studies have investigated the frequency of viral infections in pediatric patients with malignancies and shown that common pathogens (VZV, CMV, EBV, and HSV) were primarily identified as the cause of death. Moschovi et al. showed that many of the viruses leading to fatal outcomes belonged to the herpes virus family [8].

According to these findings, Buus-Gehrig et al. reviewed the published data on symptomatic infection from CMV, HSV, VZV, parvovirus B19, and adenovirus in pediatric acute leukemia and reported these events in 68 children of whom 16 patients, mostly young children, died from the infection mainly during the induction or maintenance therapy [9]. Among these viruses, no mention was made of RSV, even though it is one of the main causes of respiratory infections in children. It should be specified that some viral infections could not be diagnosed because of a lack of awareness due to an atypical presentation that might have led to an underestimation of these infections [9]. Indeed, in viral infections, some symptoms might overlap; thus, there should be a microbiological diagnosis, and strict isolation is recommended for immunosuppressed patients hospitalized during seasonal outbreaks of viruses.

One limitation of this study is the lack of comparative data; however, this is due to the limited number of similar cases reported in the literature to date. In fact, only a few cases have described the RSV infection in leukemia. Hakim et al. studied 223 children with newly diagnosed ALL at St. Jude Children’s Research Hospital, analyzing respiratory specimens, such as nasopharyngeal swab or wash, tracheal aspirate, and/or bronchoalveolar lavage [10]. The 43% of these patients who presented with viral acute respiratory illness showed that influenza was the most common virus (38%) followed by RSV with 33%. The study further indicated an infection-related mortality rate of 0.7% [10]. They confirmed a seasonal distribution with a peak in February and showed that the major prevalence of these infections occurred in neutropenic patients receiving induction chemotherapy, as seen in three of the cases in the present study [10]. The largest study on immunocompromised pediatric patients with RSV infection was conducted by Ross et al. [11]. They studied 391 immunocompromised patients with RSV infection, of whom 56% suffered from ALL. The ALL patients showed the highest number of severe RSV lower airways infections. As seen in the report, patients with lymphopenia at the onset of RSV infection showed increased risk of death [11].

In the present work, we have described four cases diagnosed in our center with different clinical manifestations. Three patients had only respiratory involvement: two of them presented with mild symptoms and one evolved into severe respiratory failure. We assumed that the evolution in respiratory distress could be related to the specific biological features of the latter patient who might be predisposed to respiratory infections, as mentioned in his remote medical history and as observed during his first hospitalization. Relevant studies have described new concepts and principles of genetic predisposition to viral infections in humans focusing mainly on defects related to innate immune responses or constitutive immune mechanisms [12]. Thus, the dysregulated antiviral immunity could be a consequence of a general immune dysregulation manifested with ALL. Furthermore, neutrophil–lymphocyte ratio and personal history of atopy become variables of a new nomogram for predicting the risk of severe acute lower respiratory tract infection; thus, they should be noted at the time of diagnosis [13]. The first case reported here involved a child with B-ALL who developed an RSV infection during induction therapy, but the diagnosis was made after the child’s death because of the extrapulmonary involvement of the RSV infection. An atypical presentation could lead to a misdiagnosis, and only a few cases have been described. Al-Maskari et al. reported the case of a six-year-old girl who developed an acute necrotising encephalopathy during an RSV infection [14]. She was initially treated with intravenous methylprednisolone and intravenous immunoglobulin and then received five sessions of plasmapheresis with an important clinical improvement and no neurological deficits [14]. Miura et al. described a case of an immunocompromised child with ALL, presenting during the maintenance phase, who displayed a high amount of RSV type B RNA in tracheal aspirates, and a serum sample was described [15]. The patient developed severe myocarditis caused by the RSV infection, which was diagnosed by abnormal findings of cardiac echography and increased biomarkers for myocardial damage. Due to the severity of the clinical picture, she was treated in the intensive care unit for 13 days [15].

Some authors described children with RSV infection and liver impairment, with both mild and severe involvement, as seen in our case. Kirin et al. presented a 13-month-old child with clinical signs and symptoms of RSV airway infection, confirmed by nasal lavage fluid and associated with elevated liver enzymes [16]. Conversely, Bakalli showed a one-month-old infant with an RSV infection documented by real-time reverse transcriptase PCR assay, manifested with sepsis and severe liver injury leading to liver failure and admission into the pediatric intensive care unit [17].

The pathophysiologic mechanisms involved in the RSV infection are related to the release of inflammatory cytokines and chemokines that affect the recruitment of inflammatory cells from the bloodstream into infected tissues and could involve not only the respiratory system but also other organs. Furthermore, the development of ALL has been related to an incorrect training of the immune system, due to decreased infection exposure [18]. A poor immune surveillance would promote the onset of ALL, and at the same time could be accompanied by a low defense against viral infections. The outcome can be modulated by the viral pathogen as well as many host factors, such as genetic susceptibility that can influence the efficiency of antiviral defenses and virus-mediated injury, justifying the individual variability in the severity of infection [19].

Current treatment options against RSV include mainly supportive care, such as supplemental oxygen, adequate hydration, and ventilation for airway impairment. As antiviral therapy, the use of oral or aerosolized ribavirin in hematological subjects was reviewed by Tejada et al. They found that mortality was significantly lower in hematological subjects and that ribavirin was well tolerated, even though the routine use of aerosolized ribavirin is still hampered by the cost and difficulty of administration [20].

Prevention strategies rely on the use of a humanized monoclonal antibody (palivizumab), such as passive prophylaxis, administered only in early preterm newborns and those with underlying cardiopulmonary diseases [21]. A promising strategy for the future could be based on new long-active monoclonal antibodies that seem to guarantee significant protection against RSV for at least five months and could be available to all infants regardless of variables, such as gestational age, presence of comorbidities, and maturity of the immune system [22]. Indeed, in July 2023, the approval of nirsevimab, the first long-acting monoclonal antibody, occurred. This antibody is used in the prevention of RSV disease and can be administered to all infants under eight months of age and for children from 8 to 19 months of age who are at increased risk for severe RSV disease [23,24]. Moreover, nirsevimab proved to be efficient in preventing the use of medical care in-term and pre-term infants, and successful in reducing hospitalization [25]. Another monoclonal antibody is clesrovimab, which was recently evaluated in a phase 2b/3 trial (NCT04767373), which showed the efficacy of decreasing RSV infection in healthy pre- and full-term infants [26]. A further effective long-acting antibody, trinomab, is currently under investigation in clinical trials for RSV prevention in infants [27].

There are also a few vaccine-based treatments for preventing pediatric RSV infections that include particle-based, vector-based, live attenuated or chimeric, subunit, and mRNA vaccines. In particular, thanks to the great success of mRNA vaccines in COVID-19, mRNA vaccines have been rapidly produced, thus demonstrating important results and acceptable safety profiles. In fact, an investigational single-dose of the mRNA-1345 vaccine against RSV in adults over 60 years of age was developed. Therefore, these vaccines might represent a new and effective preventative measure against RSV [28].

## 4. Conclusions

RSV infection could cause dramatic consequences, especially in immunocompromised children, as we reported in our case study. Some viral infections could not be diagnosed because of a lack of awareness related to an atypical presentation. Indeed, in viral infections some symptoms might overlap and might lead to an underestimation of infection. Thus, we recommend microbiological diagnosis and strict isolation mostly for immunosuppressed patients hospitalized during seasonal outbreaks of viruses. It is important to remain cognizant of the occurrence of this infection and of its atypical manifestations in order to detect and treat the infection early, thus decreasing the risk of morbidity and mortality.

## Figures and Tables

**Figure 1 pediatrrep-17-00095-f001:**
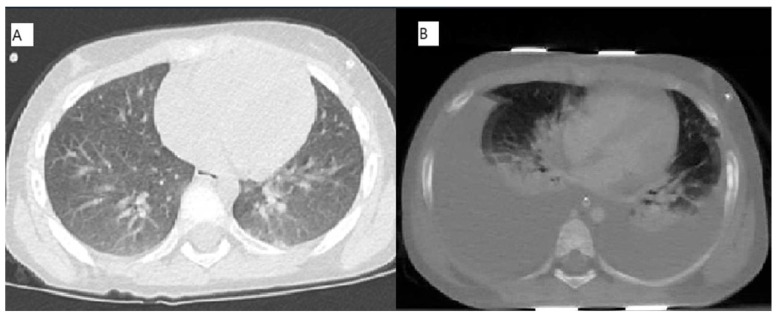
(**A**): CT-Chest performed on day 22 of induction phase; (**B**): CT-Abdomen/Chest performed on day 24 of induction phase.

**Figure 2 pediatrrep-17-00095-f002:**
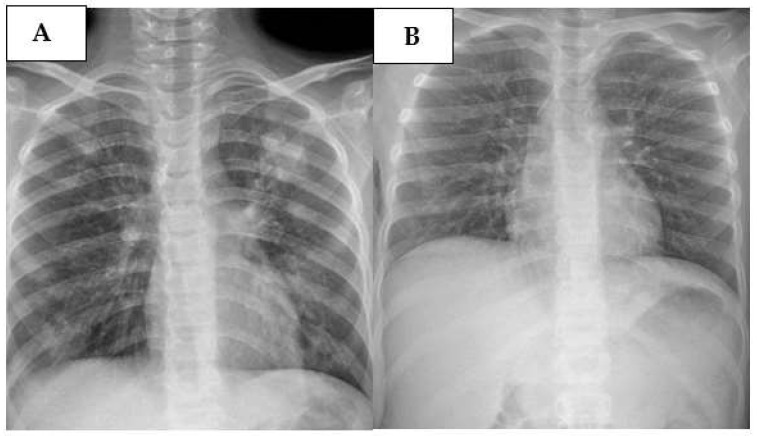
(**A**): Chest X-ray performed at the onset of the infection, at the bedside; (**B**): chest X-ray performed after 8 days of treatment.

**Figure 3 pediatrrep-17-00095-f003:**
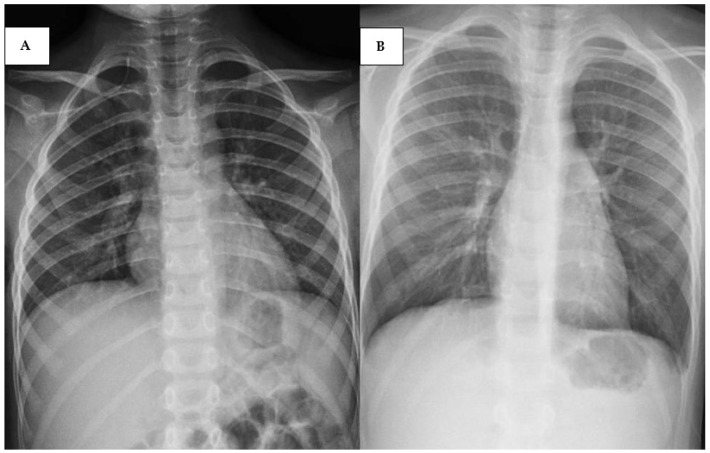
(**A**) Chest X-ray performed on patient 3 showing a slight accentuation of the peribronchovascular texture; (**B**) chest X-ray performed on patient 4 showing similar features of (**A**).

**Table 1 pediatrrep-17-00095-t001:** Laboratory findings of Case 1 (UPN 1069066) during induction.

	d 0	d + 8 ^	d + 12 *	d + 15	d + 18 ^	d + 19	d + 20	d + 21	d + 22 (8:00 a.m.)	d + 22 (4:00 p.m.)	d + 23 (8:00 a.m.)	d + 23 (4:00 p.m.)	d + 24 (Exitus)
WBC (10^3^/µL)	12.9	2.2	1.3	1.4	0.540	0.620	0.420	0.570	0.570		0.440	0.540	0.900
Hb (g/dL)	7	8.9	11.6	10.9	8.8	8.6	8.2	8.1	7.8		6.5 *	9.2	7.9
PLT (10^3^/µL)	61	46	69	56	95	104	96	95	86		23 *	24	52
CRP (mg/L)	67		14	40	12	8.22	5.07	6.25	9.96		14.36		7.69
PCT (mg/L)				0.7	0.21		0.16		1.06		0.62		0.52
AST (U/L)	10	170	55	48	80	125	181	603	2643	3531	5884		5169
ALT (U/L)						86			2857	4354	7735		6737
Bil. tot (mg/dl)	0.27	0.43	0.38		0.59				0.67	0.55	1.39		2.26
PT (s)									1.63	2.04	3.07	3.82	not measurable
PTT (s)									44	52	57	69	not measurable
Fibrinogen (mg/dL)									71	41	48 *	67 *	183
ATIII (%)									59 *	98	63		32
Pancreatic Amylase (U/L)			15			19			27		26		27
LDH (U/L)	374		160			248							7863

Table 1 Legend: d: day; WBC: White Blood Count; Hb. Hemoglobin; PLT: platelets; CRP: c-reactive-protein; PCT: procalcitonin; AST: aspartate aminotransferase; ALT: alanine aminotransferase; PT: prothrombin time; PTT: partial thromboplastin time; AT: antithrombin; LDH: lactate dehydrogenase; ^ d + 8 and d + 18: administration of Vincristine and Daunorubicin. * d + 12: administration of Peg-Asparaginase.

## Data Availability

The data presented in this study are available on request from the corresponding author due to privacy restrictions.

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
