# Peer review of "Respiratory Syncytial Virus Infection in Children with Acute Lymphoblastic Leukemia (ALL): A Contemporary Emerging and Struggling Clinical Event"

_pediatrrep, 2025, doi:10.3390/pediatric17050095_

Round 1

Reviewer 1 Report

Comments and Suggestions for Authors

The article makes a valuable contribution to the topic of severe viral complications in children with acute lymphoblastic leukemia (ALL), focusing on RSV infections – an underreported threat in this patient population. The work is well documented clinically, supported by a solid literature background, and its message holds significant relevance for clinical practice and diagnostic awareness.

The topic addressed is highly important and precisely presented, with detailed case descriptions enriched by substantial clinical and diagnostic context. Additionally, the literature review is thorough and valuable, and the conclusion is concise, insightful, and practical.

My suggestions for improvement relate to the following points:

  1. The lack of comparative data, although this is somewhat justified by the pioneering nature of the publication.
  2. The methodology of case selection is unclear. Please specify the inclusion and exclusion criteria.
  3. The discussion on available preventive strategies could be expanded (e.g., beyond palivizumab).
  4. Many sentences require language editing due to issues with clarity and style.

Author Response

Dear Reviewer,

thank you for your comments and advices. On behalf of the authors, I reply point-by-points:

Comment 1: The lack of comparative data, although this is somewhat justified by the pioneering nature of the publication.

Response 1: A limitation of our study is the lack of comparative data; however, this is due to the limited number of similar cases reported in the literature to date. In fact, only in few cases the RSV infection in leukaemia is described. 

This change can be found in page 7, paragraph 3 and lines 208,209,210.

Comment 2: The methodology of case selection is unclear. Please specify the inclusion and exclusion criteria.

Response 2: We included patients who developed RSV infection associated with ALL diagnosed at our centre from COVID-era 2020 up to now.

We included this modification in page 2 , paragraph 1 and lines 56-57.

Comment 3: The discussion on available preventive strategies could be expanded (e.g., beyond palivizumab).

Response 3: We have accordingly revised the manuscript to expand this point. Please check in page 8 , paragraph 3 and lines 283-296.

Comment 4: Many sentences require language editing due to issues with clarity and style.

Response 3: The text was edited by American Manuscripts.

Reviewer 2 Report

Comments and Suggestions for Authors

The study presented by Arrabito and colleagues on respiratory syncytial virus (RSV) in children with acute lymphoblastic leukemia is of considerable interest, as it reports four clinical cases, one of which, unfortunately, resulted in a fatal outcome. Overall, this study is highly relevant, as the authors highlight the importance of recognizing both typical and atypical manifestations of RSV infection to ensure early detection and reduce the risk of morbidity and mortality.

Some information is lacking and should be added if possible:

I recommend expanding the introduction by providing a more detailed description of RSV, its pathogenicity and typical symptoms of infection to better contextualize the topic.

Additionally, the bibliography should be enhanced to strengthen the theoretical framework and support the analysis presented.

Case 1- lines 59-65 on the 21st and 22nd day: Given the patient's age and overall condition, why were respiratory infections other than SARS-CoV-2 not immediately considered? If additional microbiological tests beyond the SARS-CoV-2 assay were performed, it is recommended that they be explicitly reported.

Case 2- line 115: There is a lack of information regarding the analytical methods used (I am referring to the PCR). I recommend pointing them out. Moreover, was the PCR positivity for the RSV monitored during hospitalization? If so, after how many days did the patient test negative for RSV? Please point it out.

Case 3 and 4- lines 152-153: What film array panel was employed? You should indicate this. Also, I suggest showing the results of the entire respiratory Biofire- Film- Array panel. As previously requested, was the positivity for the RSV monitored during hospitalization? If so, after how many days did the patients test negative for RSV? Please point it out.

Comments on the Quality of English Language

While the manuscript is generally well written, it contains several syntactic and stylistic issues that should be addressed to improve clarity and maintain an appropriate academic tone. A minor revision of the English language is advised.

Author Response

Dear Reviewer,

thank you for your comments and advices. On behalf of the authors, I reply point-by-points:

Comments 1: I recommend expanding the introduction by providing a more detailed description of RSV, its pathogenicity and typical symptoms of infection to better contextualize the topic. The bibliography should be enhanced to strengthen the theoretical framework and support the analysis presented.

Response 1: We have accordingly modified the manuscript to expand this point. Please check in page 1-2 , paragraph 1 and lines 36-47. The bibliography was enhanced as requested in lines 314-320, page 9.

Comments 2: Case 1- lines 59-65 on the 21st and 22nd day: Given the patient's age and overall condition, why were respiratory infections other than SARS-CoV-2 not immediately considered? If additional microbiological tests beyond the SARS-CoV-2 assay were performed, it is recommended that they be explicitly reported.

Response 2: Right after the performance of Broncho-alveolar lavage, we tested many viruses as A & B influence, Sars-Cov-2 and RSV. 

Please check in the text in page 2, paragraph 2.1, lines 79-80

Comments 3: Case 2- line 115: There is a lack of information regarding the analytical methods used (I am referring to the PCR). I recommend pointing them out. Moreover, was the PCR positivity for the RSV monitored during hospitalization? If so, after how many days did the patient test negative for RSV? Please point it out.

Response 3: In particular, respiratory pathogens were detected using the BioFire® FilmArray® Respiratory Panel, a multiplex PCR-based system that simultaneously identifies multiple respiratory viruses and bacteria from a single nasopharyngeal swab. The system provides automated extraction, amplification, and detection, and delivers results within approximately one hour.

This modification can be found in page 5 , paragraph 2.2 and lines 136-140.

No follow-up PCR testing for RSV was performed during hospitalization, because clinical conditions rapidly ameliorated.

Comments 4: Case 3 and 4- lines 152-153: What film array panel was employed? You should indicate this. Also, I suggest showing the results of the entire respiratory Biofire- Film- Array panel. As previously requested, was the positivity for the RSV monitored during hospitalization? If so, after how many days did the patients test negative for RSV? Please point it out.

Response 4: They showed positivity for RVS in the BioFire® FilmArray® Respiratory Panel (encompassing many viruses as Adenovirus, Coronavirus, Rhinovirus, Enterovirus, Metapneumovirus, Influenza, Parainfluenza, RSV) 

Please check the text in the page 6, paragraph 2.3, lines 175-177

As the former case no follow-up PCR testing for RSV was performed during hospitalization, because clinical symptoms were mild and general conditions rapidly ameliorated.

Reviewer 3 Report

Comments and Suggestions for Authors

              This is a report of a single-center case series of four pediatric ALL patients who developed RSV infection. The purpose of the paper seems to be that pediatric patients with immunodeficiency such as ALL are prone to severe RSV infections, so it is important to detect the disease early while paying attention to symptoms other than respiratory symptoms.

              The clinical significance of respiratory viral infections, including RSV, in immunocompromised patients, including those with malignant hematological diseases, has been reported in various ways since 1983 (Am J Med Sci 1983; 285(3):28-33). Among them are relatively large retrospective observational studies, as cited in the paper. With the COVID-19 pandemic, comprehensive PCR testing for respiratory viral infections, including RSV, has rapidly expanded from clinical research-level testing to general medical practice. In light of this, the main point of this paper is considered to be outdated. The current clinical challenge is no longer early detection, but rather clinical consideration of prevention and novel therapeutic interventions for immunocompromised patients with RSV infection. Several systematic reviews on prevention and treatment have also been reported. (Cochrane Database Syst Rev  2004 Oct 18;(4):CD000181, Cochrane Database Syst Rev 2007 Jan 24;(1):CD000181, Cochrane Database Syst Rev (IF: 9.27; Q1). 2010 May 12;2010(5):CD000181, MedComm  2024 Nov 21;5(12):e70016, Clin Microbiol Infect. 2023;29(10):1272-1279) As the case series report, it is not considered novel and is not deemed worthy of reporting.

              I am afraid to say that it is determined that the content of this paper has no clinical value based on the above points.

Author Response

Dear Reviewer,

thank you for your comment.

Comment: The clinical significance of respiratory viral infections, including RSV, in immunocompromised patients, including those with malignant hematological diseases, has been reported in various ways since 1983 (Am J Med Sci 1983; 285(3):28-33). Among them are relatively large retrospective observational studies, as cited in the paper. With the COVID-19 pandemic, comprehensive PCR testing for respiratory viral infections, including RSV, has rapidly expanded from clinical research-level testing to general medical practice. In light of this, the main point of this paper is considered to be outdated. The current clinical challenge is no longer early detection, but rather clinical consideration of prevention and novel therapeutic interventions for immunocompromised patients with RSV infection. Several systematic reviews on prevention and treatment have also been reported. (Cochrane Database Syst Rev  2004 Oct 18;(4):CD000181, Cochrane Database Syst Rev 2007 Jan 24;(1):CD000181, Cochrane Database Syst Rev (IF: 9.27; Q1). 2010 May 12;2010(5):CD000181, MedComm  2024 Nov 21;5(12):e70016, Clin Microbiol Infect. 2023;29(10):1272-1279) As the case series report, it is not considered novel and is not deemed worthy of reporting.

Response:  We respectfully acknowledge the concern that the clinical importance of RSV infections in immunocompromised patients has been widely studied, particularly in the context of hematological malignancies. However, we would like to clarify the purpose and contribution of our manuscript.

Our intention was not to present RSV as a newly discovered risk in this population, but to highlight the real-world clinical course and implications of RSV infection in children with acute lymphoblastic leukemia (ALL), based on recent and direct observations. Despite the widespread availability of multiplex PCR after the COVID-19 pandemic, there is still limited literature describing specific case presentations, clinical decisions, and outcomes in pediatric ALL patients with RSV infection, particularly in the post-COVID diagnostic landscape.

We believe that this case series offers educational value, contributes to the clinical understanding of RSV impact in this vulnerable subgroup, and may be of interest to clinicians managing similar patients. While we agree that prevention and therapy are the critical areas of research and intervention, documentation of clinical cases remains relevant for the medical community, especially in those situations where evidence-based guidelines are still evolving. 

Round 2

Reviewer 3 Report

Comments and Suggestions for Authors

I would like to thank you for giving me the opportunity to review your valuable revised paper.

As I mentioned in my comments on the first edition, unfortunately, I cannot provide a clinically meaningful assessment of this case series report. Unfortunately, my evaluation will be the same as before.